# Characteristics of Plastic Waste Processing in the Modern Recycling Plant Operating in Poland

Elżbieta Szostak [1,*], Piotr Duda [2], Andrzej Duda [2], Natalia Górska [1], Arkadiusz Fenicki [3] and Patryk Molski [3]

[1] Faculty of Chemistry, Jagiellonian University, ul. Gronostajowa 2, 30-387 Kraków, Poland; natalia.gorska@uj.edu.pl
[2] Faculty of Mechanical Engineering, Cracow University of Technology, Al. Jana Pawła II 37, 31-864 Kraków, Poland; pduda@mech.pk.edu.pl (P.D.); andrzej.duda@pk.edu.pl (A.D.)
[3] GreenTech Polska S.A., Al. Prymasa Tysiąclecia 46-205, 01-242 Warszawa, Poland; a.fenicki@greentechpolska.pl (A.F.); patryk.molski@greentechpolska.pl (P.M.)
* Correspondence: elzbieta.szostak@uj.edu.pl; Tel.: +48-126-862-254

**Abstract:** Although Poland is one of the leading recipients of the waste stream in the European Union (EU), it is at the same time below the average in terms of efficiency of their use/utilization. The adopted technological solutions cause waste processing rates to be relatively low in Poland. As a result, the report of the Early Warning and Response System (EWRS) of the EU indicated Poland as one of the 14 countries of the EU which are at risk in terms of possibility of achieving 50% recycling of waste. This article discusses the implemented technological solutions, and shows the profitability of the investment and the values of the process heat demand both for extractor and reactor. The experimental part analyzed the composition of the input and output of the process and compared it to the required fuel specifications. Attention was drawn to the need to improve the recycling process in order to increase the quality of manufactured fuel components. As potential ways of solving the problem of low fuel quality, cleaning the sorted reaction mass from solid particles and extending the technological line with a distillation column have been proposed. The recommended direction of improvement of the technology is also the optimization of the process of the reactor's purification and removal of contaminants.

**Keywords:** waste management; pyrolysis of plastics; recycling of polyolefin plastics on a full industrial scale



## 1. Introduction

Proper waste management is one of the key elements of modern society. Sensible management of processed materials is based on the implementation of infrastructure for the controlled storage and processing of waste as well as the enforcement of related legislation. According to the data of the Chief Inspectorate of Environmental Protection, Poland is one of the leading recipients of the waste stream in the European Union (EU). Over the past four years (2016–2019), as much as 1.5 million tons of processed waste were transported to Poland mainly from Germany, Italy, Austria, Denmark, Slovenia, and Great Britain [1]. At the same time, as indicated by the data presented in the Country Report Poland in the Environmental Implementation Review 2019 of the European Commission [2], the prevailing method of waste management in Poland in the years 2010–2017 was landfilling (Figure 1).

Waste storage over a long period of time carries the risk of serious health and environmental problems [3]. It promotes groundwater pollution [4] and emission of greenhouse gases (GHGs), and also creates a risk of occurrence of landfill fires and explosions [5]. This can be proven by the degradation of the environment and sanitary problems faced by Polish residents in 2017 and 2018 as a result of illegal acts of arson fires of landfilled waste [6]. This situation prompted the Polish government to tighten up the waste management regulations at the end of 2018—Act of 20 July 2018 amending the Act of Inspection

of Environmental Protection and certain other acts, Journal of Laws of 2018, item 1479 [7]. The result of this tightened approach is a fight against pathologies in waste management. The amendment regulates the time of landfill, introduces mandatory monitoring of landfills, and allows for their continuous control. It also limits international transport of waste to those materials that can be used in a recycling process and provides for severe financial penalties for noncompliance. In turn, a regulation establishing rules on separate collection of household waste was introduced at national level (Journal of Laws of 2019, item 2028) [8]. In the future, these regulations may increase not only the quality of recycled materials, but also their economic value.

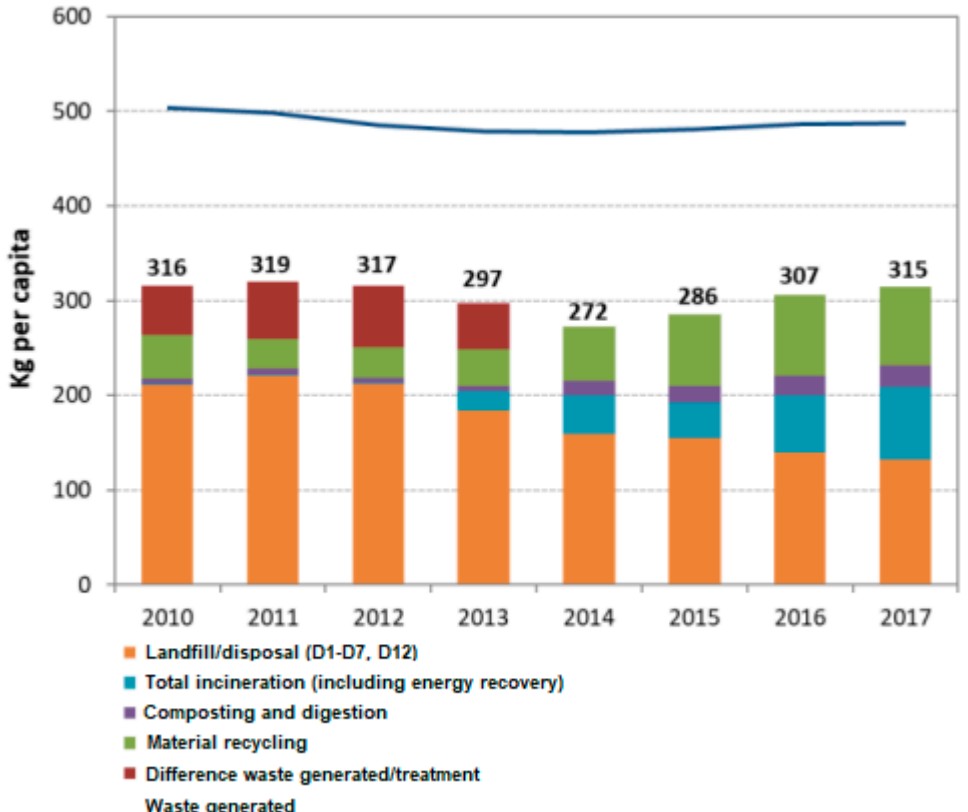

**Figure 1.** Ways for plastic waste management in Poland between 2010 and 2017 adapted from the European Commission, Eurostat, Municipal waste by waste operations.

Although, from a legal point of view, the amendments made will tighten the regulations on landfill, they will not improve the waste management process. Proper waste management depends not only on the legal situation, but also on the technological solutions adopted. According to 10 indicators in the framework of monitoring the closed-loop economy, Poland is below the EU average in terms of material resource efficiency for the production of wealth in a closed (secondary) circuit (10.2% in 2016, compared to the EU average of 11.7%) (European Commission, Eurostat, Resource productivity). Although Polish residents produce less municipal waste than the EU average (e.g., in 2017 it was 315 kg/inhabitant from Poland and 487 kg/inhabitant from the EU), only 34% of it was recycled. At the same time, the EU average of waste treatment reached 46% (European Commission, Eurostat, Municipal waste by waste operations). The high rate of import of waste materials and, at the same time, the low rate of their processing make the European Commission's early warning report indicate Poland as one of the 14 EU countries in which the achievement of the target of 50% recycling of waste by 2020 may be threatened (European Commission report on the implementation of the EU waste legislation including the early warning report for Member States at risk of missing the 2020 preparation for

re-use/recycling target on municipal waste, COM(2018) 656 and SWD(2018) 422). This situation is unusual inasmuch as the number of companies operating in Poland converting plastic waste into new materials is one of the highest in the whole of Europe (see Figure 2).

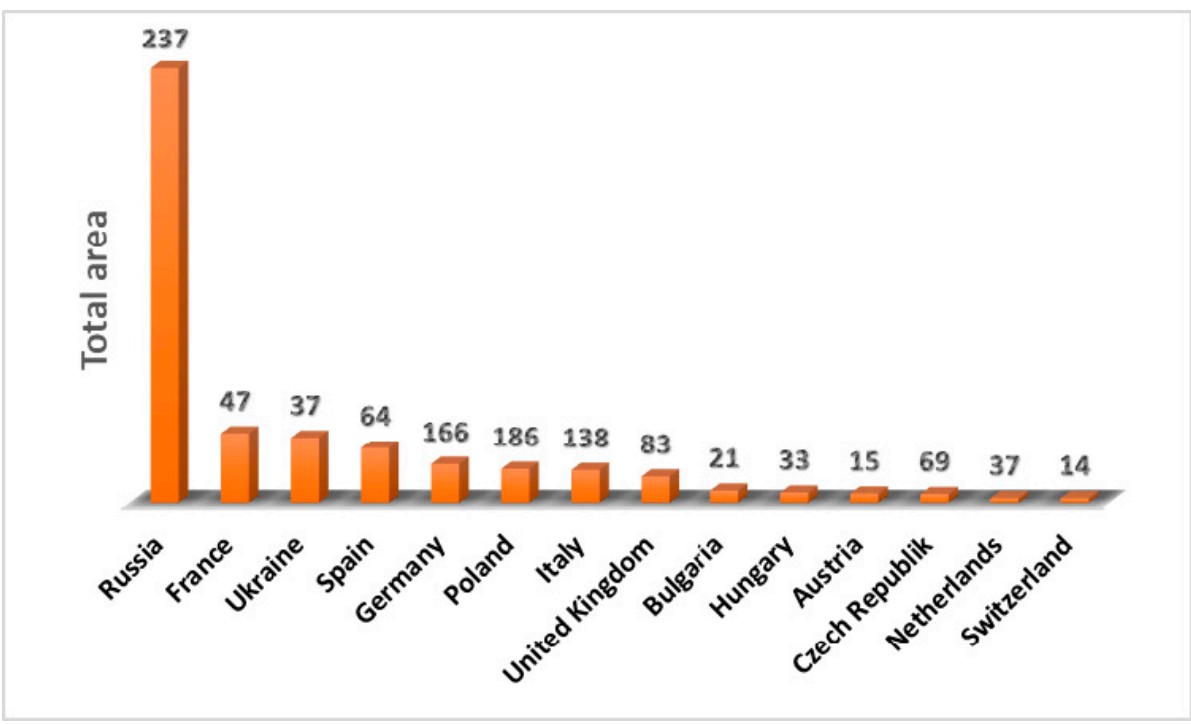

**Figure 2.** The size of European countries vs. the number of plastic recycling plants according to ENF [9].

The aim of this work was to analyze in detail the plastic processing line in the Packaging Waste Recycling Plant in Toruń, owned by GreenTech Polska S.A., one of the companies entering the market and operating in the recycling industry in Poland. Interestingly, the basis of the company's activity is pyrolytic process conducted in a noncatalytic and simplified technology (no rectifying column or at least distillation one, hydrorefining). The research was carried out from the early stage of material collection through their segregation, processing, and analysis of the manufactured end products. The results of the research will allow assessing the advantages and disadvantages of the noncatalytic recycling process of plastics as well as evaluating parameters that should be taken into account in order to increase recycling efficiency and the plastics processing rate in Poland. The quality of materials produced in a noncatalytic pyrolysis process will also be assessed.

## 2. Material and Methods

### 2.1. Characteristics of the Subject of Research

2.1.1. Production Plant

Packaging Waste Recycling Plant, whose processing lines were investigated in this study, occupies the area of the industrial economic zone in Toruń. The following industrial buildings are located on the site:

- production-storage building with equipment for storing, sorting, and compacting of segregated waste brought into the plant
- control room building with control node for the technological installation
- building of the transformer station with voltage ratio of 400/6300 [V] with the main power switchboard of the plant
- shed of pyrolysis node with technological installation for thermal processing of mixed packaging and multilayered waste
- two sets of power generators

  − liquefied petroleum gas (propane) and liquid fuels supply station for technological purposes (i.e., heating the reactor during startup or operation of the plant)
  − two underground storage tanks for liquefied petroleum gas (propane)
  − two underground storage tanks for liquid products forming in technological process
  − truck tanker loading station with liquid products forming in technological process

By decision of the competent authority, it has been recognized that the plant meets all the construction, sanitary, and environmental requirements and does not present a major accident hazard.

### 2.1.2. Processed Raw Materials

The basis of the plant's activity is sorting, shredding, and thermal processing of waste coming in most cases from municipal waste. Characteristics of these wastes are presented in Table 1.

**Table 1.** Types of wastes processed by the company (the waste code numbers assigned according to Journal of Laws of the Republic of Poland of 2020, item 10).

| No. | Waste Code | Type of Waste |
|-----|-----------|---------------|
| 1. | 03 03 07 | Mechanically separated rejects from processing of waste paper and cardboard |
| 2. | 07 02 13 | Plastic waste |
| 3. | 15 01 02 | Plastic packaging |
| 4. | 15 01 05 | Multimaterial packaging (tetra packs) |
| 5. | 15 01 06 | Mixed packaging waste |
| 6. | 16 01 99 | Other waste not specified |
| 7. | 17 02 03 | Plastics |
| 8. | 19 12 04 | Plastics and rubber |
| 9. | 19 12 12 | Other wastes (including mixed substances and objects) obtained from mechanical waste treatment other than mentioned in 19 12 11 |
| 10. | 20 01 39 | Plastics |
| 11. | 20 01 99 | Other fractions selectively collected not mentioned above |

The waste is delivered to the plant site in a baled form (bales with a volume of 1 m$^3$ and a weight from 200 to 500 kg) or loose-loaded mass of bulk density from 40 to 60 kg/m$^3$ (Figure 3).

In the initial stage, the waste stream is shredded (Figure 4a). Impurities are removed, mineral, metal, and glass fractions and waste paper are separated (Figure 4b). Initially, the separation takes place manually, and then with the help of a disc screen and electromagnetic, eddy current and optoelectronic separators. The residual waste stream is cleaned of materials that are undesirable, taking into consideration the designed pyrolysis equipment. At this stage, PVC, PET, and multimaterial packaging are separated (Figure 4c). Bonded in the process of contact homogenization (that is, melting and subsequent cooling and granulation) and dried, final waste, which is a mixture of polymers from the group of polyolefins, including high-density polyethylene (HDPE), low-density polyethylene (LDPE), polypropylene (PP), and polystyrene (PS) (Figure 4d), undergoes heat treatment in the pyrolysis installation. Types of pyrolyzed materials are identified by the optoelectronic Steinert UniSort P2800RR apparatus.

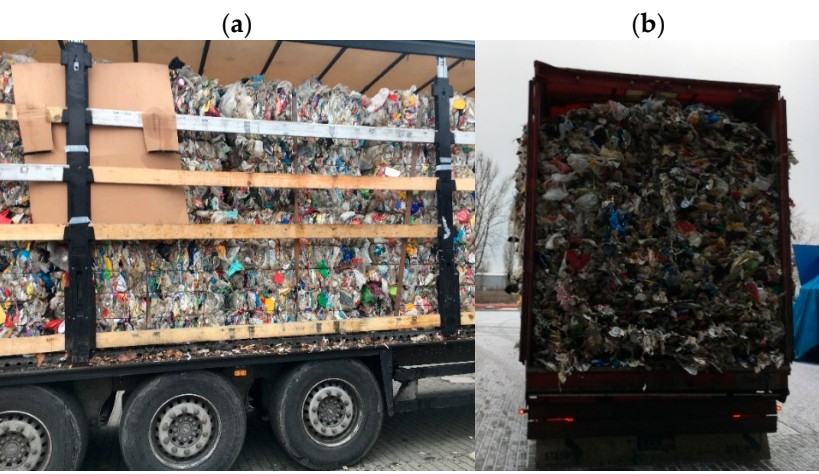

**Figure 3.** Transportation of mixed municipal waste to the plant.

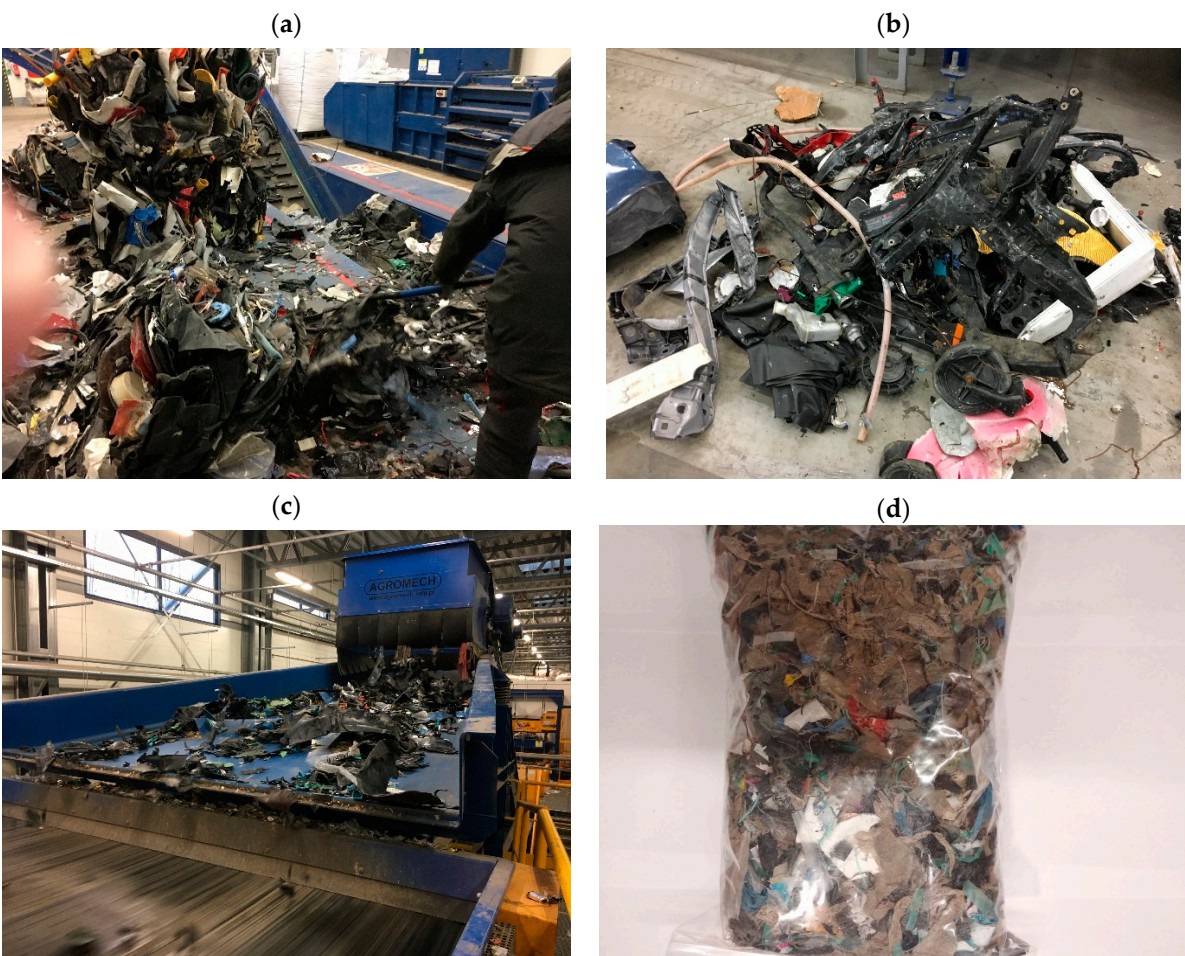

**Figure 4.** Waste sorting system: (**a**) initial shredding of mixed municipal waste, (**b**) removal of impurities and separation of mineral, metal, and glass fractions and waste paper, (**c**) cleaning of residual waste and separation of different fractions for further pyrolysis, (**d**) final waste prepared for heat treatment in the pyrolysis installation.

The plant is adapted to receive 50 tons of waste material within a day. After mechanical segregation, about 20 to 30 tons of this waste goes to the pyrolysis installation. The products obtained in the plant are sorted waste (mainly paper, cardboard, cellulose prod-

ucts, nonferrous metals, and selected plastics such as PET packaging), liquid, and gaseous pyrolysis products and electricity. Minimum and maximum processing capacity (i.e., the so-called waste throughput) is equal to 1000 kg/h.

### 3. Theory

*3.1. Technological Process*

#### 3.1.1. Thermal Cracking

The idea behind the company's market activity is the technology of recycling thermoplastics from the group of polyolefins (PE, PP, and PS) aimed at recovering from them high-energy oil derivatives by pyrolysis. The installation designed in the plant is intended to carry out the pyrolysis process [10], which is known next to glycolysis [11,12], hydrolysis [12,13], aminolysis [12,14], gasification [15], and hydrogenation [16] as the plastic processing method.

Pyrolysis belongs to thermochemical recycling methods. In a broadly defined pyrolysis process, organic compounds (polymers) heated to high temperatures (350 to 900 °C) in an inert oxygen-free atmosphere decompose, resulting in creating highly calorific gaseous and liquid products (consisting of paraffins, olefins, naphthenes, and aromatic compounds) and solid waste containing inorganic residues [4,17]. The liquid fraction can be used to recover hydrocarbons in terms of gasolines ($C_4$–$C_{12}$), fuel oils ($C_{12}$–$C_{23}$), paraffins ($C_{10}$–$C_{18}$), and motor oils ($C_{23}$–$C_{40}$). The gas fraction can be used to maintain temperature of the process and to compensate for the total energy demand of the pyrolysis installation itself [18], while carbonisate (solid waste) can be used to produce activated carbon after developing the refining technology. From an environmental point of view, the advantage of pyrolysis is the reduction of $CO_2$ emission compared to landfilling and incineration as well as incineration with energy recovery. Pyrolysis products can replace fuel oil and natural gas, avoiding 30 wt% of the $CO_2$ emitted during incineration, thus reducing the carbon footprints [19,20].

#### 3.1.2. Mechanism of Thermal Decomposition

The decomposition of polymers can be explained by one of the four mechanisms proposed below or a combination of them (Figure 5) [21].

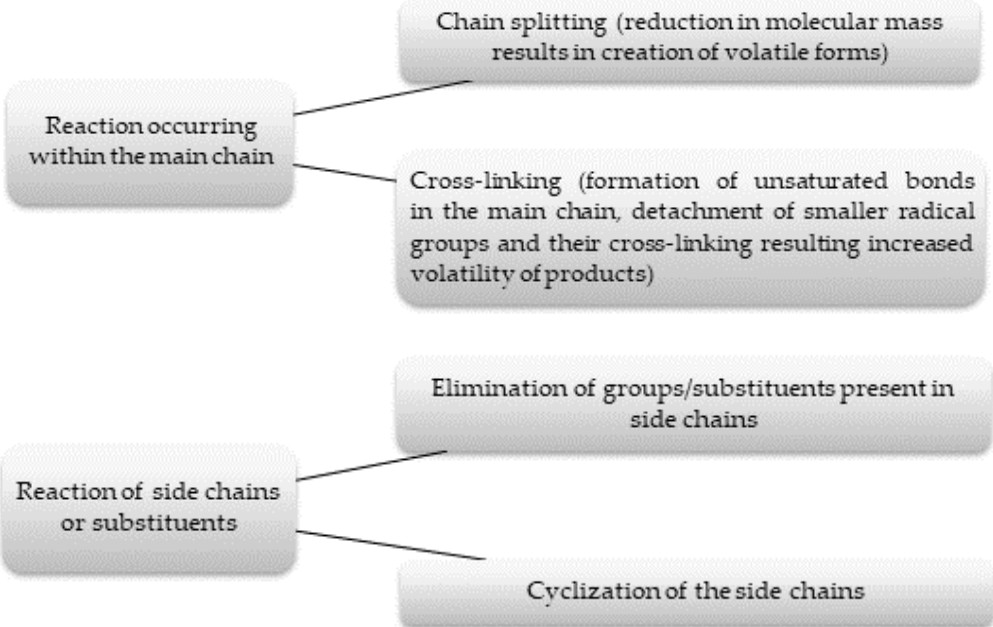

**Figure 5.** Mechanism of polymers' decomposition.

The most common mechanism of thermal decomposition of the thermoplastic polymers is the one involving breaking bonds in the main or side chain. This is a multistage free-radical reaction including initiation and chain propagation, branching, and termination. The initiation process, which results in the formation of free radicals, can take place at the ends or at random positions in the main or side polymer chain. Propagation reactions (shortening) of the polymer chains may occur as a result of the above-mentioned accidental chain scission or hydrogen atom transfer. In this second process, the radicals formed at the initial stage as a result of intramolecular transfer of hydrogen atom unzip with the formation of new radicals and low-molecular unsaturated hydrocarbons. Hydrocarbon molecules can further exchange hydrogen atoms with the radicals present in the reaction environment through intermolecular transfer, leading to further splitting of their chains and formation of new radicals and molecules with increasing level of unsaturation. Such chains of oligomers can undergo cross-linking reaction. It generally occurs as a result of removal of certain substituents and involves formation of bonds between two adjacent oligomer molecules. This process leads to generation of a chemical structure of higher molecular weight, which is less volatile. Reactions involving side chains are mainly elimination and cyclization reactions. In the elimination reactions, groups/side substituents are detached from the main chain resulting in formation of unsaturated bonds in the main chain. The radicals formed from this detachment can recombine with one another to create corresponding volatile molecules of lower molecular weight. In a cyclization reaction, two adjacent side groups react with each other. This leads to a cyclical structure. In this process, compounds with a higher carbon-to-hydrogen ratio are created.

The end of the pyrolysis reaction can be achieved due to unimolecular termination, recombination, or disproportionation. The element connecting all of the described processes is the reaction of two radicals with the formation of a permanent covalent bond.

The activation energies ($E_a$) of the pyrolysis of the HDPE, LDPE, PP, and PS types are estimated to be in the range 206–445 kJ mol$^{-1}$ [22–27], 163–303 kJ mol$^{-1}$ [22,23,26], 99–244 kJ mol$^{-1}$ [15,24,26,28–30], and 83–310 kJ mol$^{-1}$ [26,31], respectively.

### 3.1.3. Thermal Decomposition Products of Polymers

There are several models showing possible pathways for the creation of products due to thermal decomposition of polymers, which are presented in Figure 6 [32–34].

**(a)**

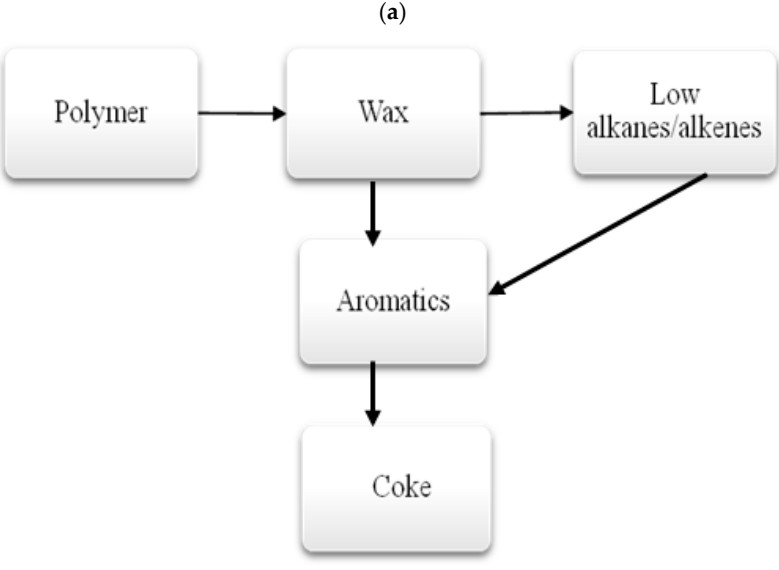

**Figure 6.** *Cont.*

(b)

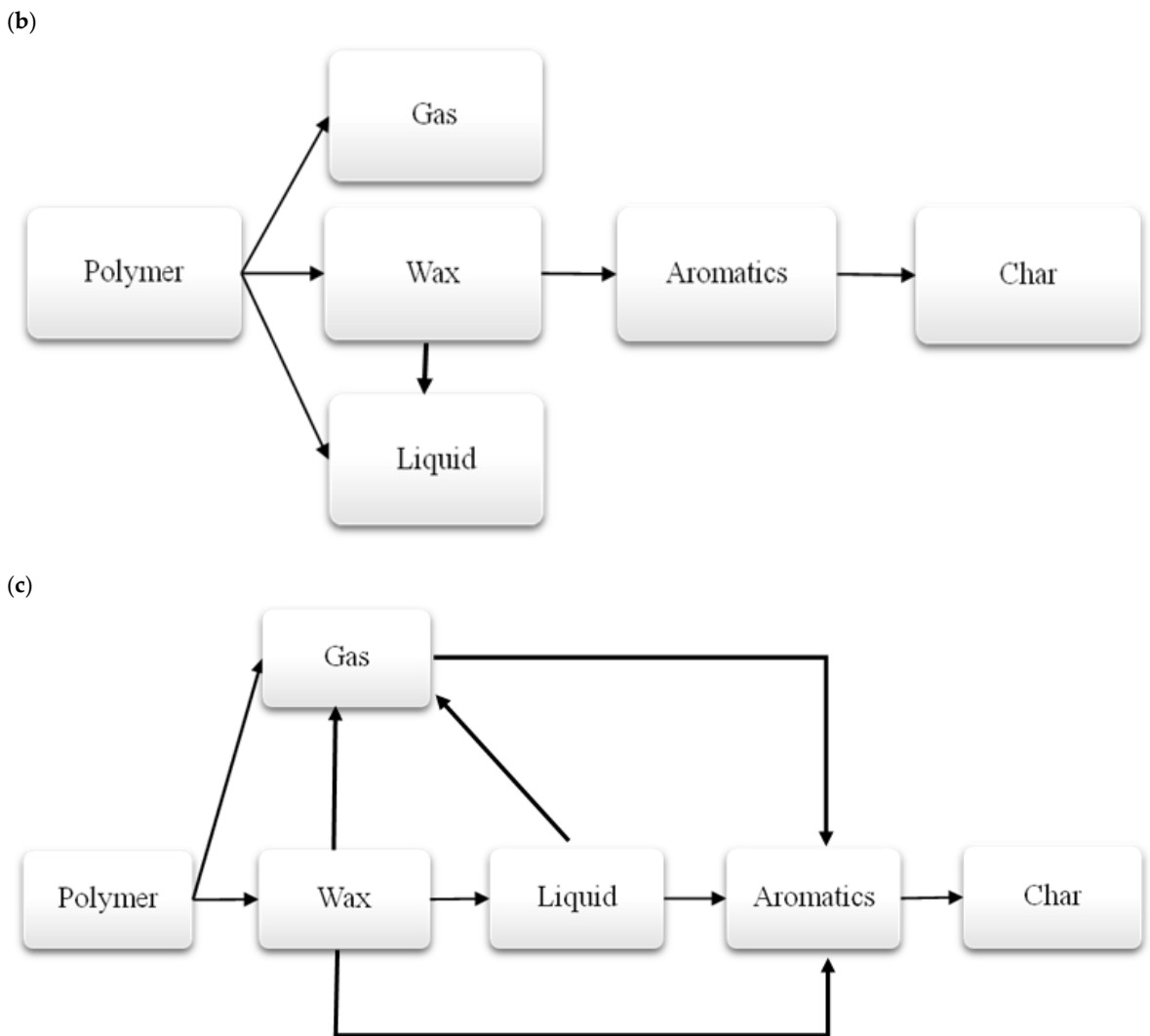

(c)

**Figure 6.** Possible pyrolytic cracking pathways.

The main objective of the technological process, on which the business model of the plant is based, is to manufacture products of properties similar to conventional excise energy products with appropriate efficiency. This parameter is greatly influenced by temperature and duration of the process, type of material to be pyrolyzed, and reaction atmosphere. It is desirable to produce motor gasoline and fuel oil, which can be achieved in the production of hydrocarbons with a boiling point of 35–185 °C (gasoline), 180–350 °C (fuel oil), and 350–538 °C (vacuum gas oil, VGO) [35]. The raw material recycled at the plant is a mixture of PE, PP, and PS.

Although the form of the final product obtained by pyrolysis is related to the grade of plastic used and the process conditions essentially, as indicated in literature, thermal cracking of LDPE and HDPE leads mainly to obtaining liquid products [36,37]. Gaseous products produced by depolymerization of PE are: ethane, ethene, and n-butane. The gaseous fraction formation efficiency increases with increasing temperature to a certain limit value [19,38,39]. High carbon content is obtained due to the presence of HDPE, whereas LDPE increases percentage of gas in the product [19,35]. Pyrolytic oil obtained on the basis of HDPE has calorific value (CV) of 42.9 MJ kg$^{-1}$ [40–43] and on the basis of LDPE 39.5 MJ kg$^{-1}$ [44]. The estimated calorific value of gaseous products of pyrolysis HDPE obtained in the temperature range 350–500 °C is equal to 50.8–52.7 MJ kg$^{-1}$ and is comparable to 55.7 MJ kg$^{-1}$ determined for pure methane gas [45].

Pyrolysis of PP is usually carried out in the temperature range of 250–500 °C. The presence of a tertiary carbon atom in the PP structure makes its hydrocarbon chain relatively easy to break, resulting in a variety of products obtained by pyrolysis. The pyrolytic oil formation efficiency under these temperature conditions reaches a value in the range of 69–93 wt% [39,43,46].

Polystyrene degrades to highly viscous, dark brown, oily liquid at 350 °C [34]. In this process about 1 wt% of solid waste (residue charring) and a small amount of gaseous hydrocarbons are also produced. An increase of temperature causes an increase of efficiency of the coking process up to 30.4 wt% at 500 °C and a decrease of efficiency of obtaining liquid fraction (up to about 67 wt%), with a small impact on the amount of gaseous products (2.50 wt%). Pyrolytic oil obtained from decomposition of PS is dominated by aromatic compounds such as: benzene, toluene, ethylbenzene, styrene, cumene, alpha-methylstyrene, diphenylpropane, and triphenylbenzene.

Studies show that degradation of mixed plastic waste occurs at temperatures lower than degradation of single polymers [19,47,48]. This phenomenon indicates a beneficial synergistic effect of pyrolysis of mixed waste and is an added value from the point of view of the recycling process conducted in the plant.

### 3.1.4. Pyrolysis Installation

Figure 7 shows a simplified scheme of the pyrolysis installation with a theoretical capacity of 1000 kg/h operating at the plant in Toruń.

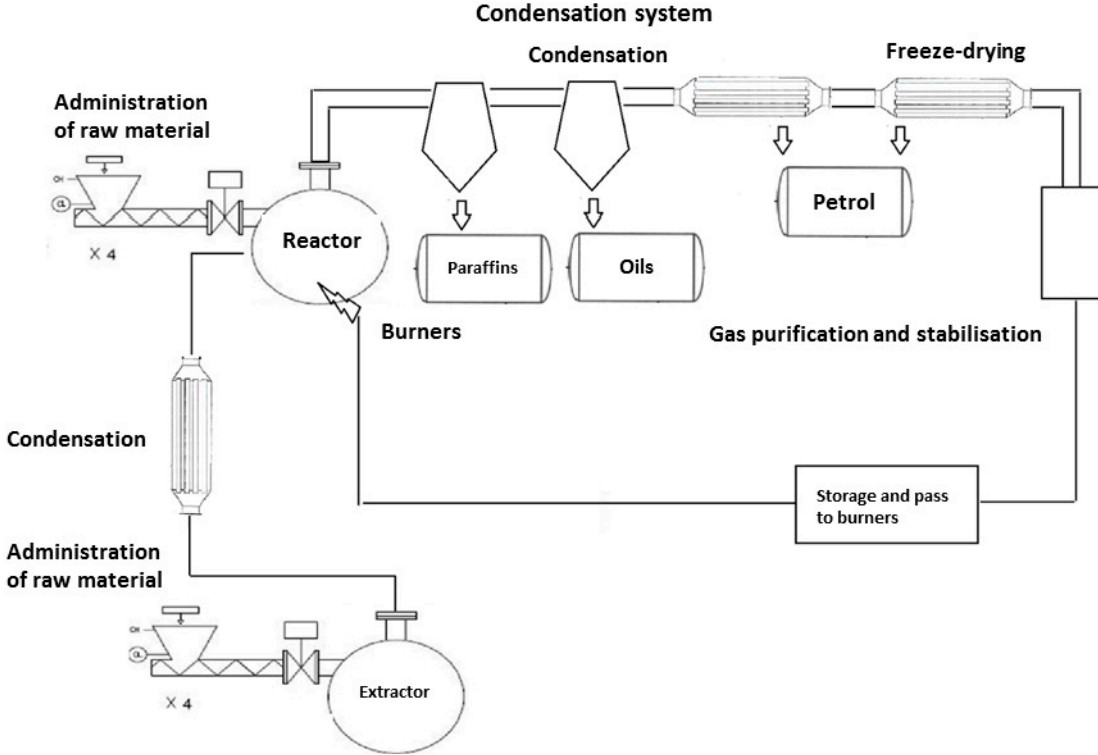

**Figure 7.** Schematic diagram of pyrolysis installation.

The installation includes:

(a) The system supplying the production line with a stream of sorted waste (PLW 600/16000 belt conveyor, T1/T4 SIGMA screw conveyors, a feeding system with presses, a system dosing raw material to the R101 reactor and R201 extractor).

(b) A system that ensures proper operation of the extractor (a solvent feed mechanism (heavy fraction from the reactor), a heat exchanger system to maintain temperature of homogenization process in the range of 180–200 °C (E202/W401 jacket-tube

exchanger in a dosed solvent-extraction mixture system, by-pass exchangers in a thermal oil-extraction mixture system), a content-mixing system (three circulating pumps P205–207 allowing extraction mass to be exchanged four times per hour), a system for liquefaction and removal of gas created in extractor (E201 heat exchanger combined with a V106 separation tank to separate aqueous and hydrocarbon fraction), a mixture extraction and cleaning system and its dosing to the reactor (P205 dosing pump with F103 filter)). In the extractor operating in the temperature range of 180–200 °C, the mass of solid waste is dissolved in liquid hydrocarbon fractions. These fractions come from the liquefaction of the reactor products and are dosed to the extractor through the pipeline equipped with a pump and automatic valves for flow control. The content of the extractor is mixed with the help of circulation pumps, which accelerates the process of dissolving solid waste and increases the heat transfer between the heating elements and the extraction mixture. After dissolving, the mixture of heavy fraction and liquid polyolefins is taken from the tank lower connector pipes and directed by pumps to the connector pipes located on the reactor side surface.

(c) A system that ensures proper operation of the reactor (a two-level reactor heating system providing the heat necessary to reach and maintain a temperature of 420 °C), set of heating inserts connected by ends to O101–108 gas burners of modulating power of 60–200 kW and an exhaust outlet to a chimney. The gas burners are fed with technological gas created at the pyrolysis stage or propane gas used at the installation start-up stage, a content-mixing system (four circulating pumps P201–204 allowing reaction mass to be exchanged eight times per hour), a reactor and reactive mixture cleaning system (thanks to the extraction system, the solid contamination of the reactor does not exceed 5%). These contaminants are periodically drawn from the bottom of the reactor and then pumped through F101–102 filters and captured by feeders of SF203–204 dryers, which direct dry matter and hydrocarbon vapors present in a filtrate to solid waste storage containers and a condensate-separation system, respectively. A condensate system for the gaseous products of pyrolysis (a system of E101-108 heat exchangers (a stream of vapors of 1000 kg and a temperature of 420 °C is directed to them every hour which, when mixed with injected hydrocarbon mixture at a correspondingly lower temperature, undergoes four-stage condensation. Non-condensing after cooling gas mixture (mainly methane, ethane, propane, and butane and traces of heavier hydrocarbons) flows through an E108 exchanger cooled with aqueous solution of propylene glycol with an inlet temperature of −5 °C). On its way, a stream of condensed vapors goes to intermediate tanks collecting different fractions (V104 tank maintained at a temperature of at least 80 °C—hydrocarbon fraction with boiling temperatures >170 °C, V107 tank maintained at a temperature of at least 60 °C —hydrocarbon fraction with boiling temperatures >120 °C and V103 tank—hydrocarbon fraction with boiling temperatures >30 °C. In the last part of the condensation path there is a V105 tank, which intercepts the gas stream leaving the last fourth stage of condensation. After purification it is used as fuel in burners feeding the technological process)).

(d) A product storage system for sale time or supply of power generator sets (M101 two-chamber tank (K1 chamber, gasoline products (CN–27 10 12 25) from V103 tank, K2 chamber, oil products (CN–27 10 19 29) from V107 chamber); M102 two-chamber tank (K3 chamber, paraffin products (CN–27 10 19 85) from V104 tank, K4 chamber, fuel mixture supplying generator sets which is a mixture of paraffin, oil and gasoline fractions); an installation ensuring the quality and possibility of product design (chambers equipped with apparatus enabling mixing and circulation of liquids, measuring and regulation of its temperature, and assessing the level and acidity of mixture); installed pump units connected with a static mixer enable preparation of fuel mixtures according to specific recipes, e.g., fuel intended for supplying generator sets) – energy product GreenOil [49].

## 4. Results and Discussion

### *4.1. Analysis of Recycling Process*

#### 4.1.1. Sorting of Waste for Pyrolysis

The basis of the designed technological process is to deliver to the system an appropriate quality of the waste stream constituting the input to the pyrolysis unit. Therefore, wooden, metal, glass, or mineral elements should not be directed to the system. The polymers subjected to the pyrolysis process should also not have organic admixtures containing Cl, N, and S elements or CO groups. The maximum moisture content and impurities in the raw material must not exceed 10% and 5% of the total load weight, respectively. Data obtained in the process of sorting waste materials with code 19 12 04 (plastics and rubbers) [50] delivered to the plant from Denmark, Sweden, Germany, and Italy are collected in Table 2.

**Table 2.** Characteristics of the waste to be recycled.

| Waste Code | Waste Classification by the Company Designation | Average Content in Weight of Waste Delivered to the Plant [%] | Standard Deviation of the Mean Value [%] |
|---|---|---|---|
| 19 12 04 | Plastics for further processing | 60.6 | 20.4 |
| 19 12 09 | Minerals (e.g., sand, stones) | 6.2 | 4.0 |
| 19 12 02 | Ferrous metals | 0.8 | 0.4 |
| 19 12 03 | Nonferrous metals | 0.8 | 0.4 |
| 19 12 12 | Other wastes (including mixed substances and objects) | 25.5 | 19.9 |

As shown by the results presented in Table 2, the waste stream delivered to the site for recycling is characterized by a great diversity. Apart from the material that is technologically significant (code 19 12 04), the waste stream includes materials that are not accepted in the designed pyrolytic process (tetra pack, paper, wood, electronic components, textile materials, footwear, PET, PCV, PUR, PA, and ABS). The fractions of waste paper, PET plastics, and ferrous and nonferrous metals (mainly Al and Cu), which find their recipients and further processors outside the plant, are mechanically separated from the waste rejected at the sorting stage. The most troublesome material obtained at the sorting stage is the rejects containing a fine fraction. Table 3 provides exemplary results of morphological tests of three samples of such rejects.

**Table 3.** Characteristics of post-sorting ballast.

| Parameter Determined | Research Method Applied | Content in Sample [wt%] | | |
|---|---|---|---|---|
| | | Sample 1 | Sample 2 | Sample 3 |
| Food waste of plant origin | | 0 | 0.09 | 0 |
| Food waste of animal origin | | 0 | 0.11 | 0 |
| Paper and cardboard waste | PN-93/Z-15006 | 30 | 29.9 | 22.20 |
| Plastic waste | | 0 | 60.4 | 74.50 |
| Textile waste | | 0 | 2.80 | 2.40 |
| Glass waste | | 0 | 0 | 0 |
| Metal waste | | 0 | 0.10 | 0.06 |
| Other organic wastes | | 0 | 3.80 | 0.09 |
| Other mineral wastes | | 0 | 2.20 | 0.01 |
| Fraction <10 mm | | 70 | 0.60 | 0.74 |

As can be seen from Table 3, the mixed waste that could not be sorted is mainly paper waste, waste made of plastics other than PE, PP, and PS, and the fine fraction of <10 mm in diameter. This waste predominantly has a calorific value exceeding 6 MJ kg$^{-1}$ (plastics 22–46 MJ kg$^{-1}$, paper and cardboard 11–26 MJ kg$^{-1}$, textiles 15–16 MJ kg$^{-1}$ ) [51]. According to the Regulation of the Minister of Economy of 16 July 2015 (Dz.U. z 2015 r., poz. 1277), such waste cannot be placed in the landfills different than hazardous and inert and therefore needs further treatment.

The most economically beneficial use of the reject fraction from the pyrolysis is to use it as an alternative source of fuel in cement plants or in waste incinerators. Alternative fuels used for cement industry must meet a number of quality requirements. Physicochemical parameters characterizing the reject fraction together with those required by cement plants are listed in Table 4.

**Table 4.** Physicochemical parameters required in cement industry and those of reject fraction obtained in the plant.

|  | Requirements of Cement Industry | Post-Sorting Ballast (Reject Fraction) | Research Method Applied |
|---|---|---|---|
| Moisture content [%] | <20 | 31.0 ± 6.2 | CEN/TS 15414-1:2010 |
| Calorific value [MJ kg$^{-1}$] | >18 | 25.21 ± 2.52 | PN-EN 15400:2011 |
| Sulfur content [%] | <0.5 | 0.06 ± 0.02 | PN-G-04584:2001 |
| Ash content [%] | <20 | 4.1 ± 0.6 | PN-EN 15403:2011 |
| Chlorine content [%] | <0.2 | 0.352 ± 0.088 | PN-EN 15408:2011 |

The post-sorting ballast obtained in the plant does not meet the quality requirements of customers from the cement industry. Therefore, the management of this fuel stream is executed by transferring it to a waste incinerator where it is used as fuel for an incinerator.

4.1.2. The Process Heat Demand

The heat demand for the extraction process is determined by the following expression:

$$QE = Q_1 + Q_2 + Q_3$$

where:

$Q_1$ is the heat required to heat the raw material to the melting point
$Q_2$ is the heat required to melt polyolefins
$Q_3$ is the heat required to evaporate the water contained in the waste

The following assumptions were made for the calculation of the heat value.
Mass fraction of the fraction in the sorted raw material:

polyolefins (soluble) 75%
other plastics (insoluble) 20%
water 5%

Other data:

waste throughput 1000 kg/h
amount of solvent supplied to extractor 2000 kg/h
input temperature of raw material 15 °C
the temperature in the extractor 140 °C
average specific heat of the solvent $c_{ps}$ = 2.81 kJ/kgK
average specific heat of water $c_{pw}$ = 4.18 kJ/kgK (at 50 °C)
average specific heat of polyolefins $c_{pP}$ = 2.3 kJ/kgK
heat of fusion of polyolefins $H_{fP}$ = 203 kJ/kg
heat of water vaporization $H_{vw}$ = 2264.67 kJ/kg (at 100 °C)

The heat demand of the extractor determined for the given parameters is:

$$Q_E = Q_1 + Q_2 + Q_3 = 80.83 \text{ kW} + 42.30 \text{ kW} + 31.45 \text{ kW} = 154.58 \text{ kW} \approx 160 \text{ kW} \ (576{,}000 \text{ kJ})$$

The heat demand of the reactor is defined by the following formula:

$$Q_R = Q_1 + Q_{2b} + Q_{2o} + Q_{2p}$$

where:

$Q_1$ is the heat required to heat and decompose the polyolefins
$Q_{2b}$ is the heat required to evaporate the gasoline fraction
$Q_{2o}$ is the heat required to evaporate the oil fraction
$Q_{2p}$ is the heat required to evaporate the paraffin fraction

The following assumptions were made for the calculation of the heat value.
Mass fraction share after the cracking process:

gas fraction 10%
gasoline fraction 10%
oil fraction 40%
paraffin fraction 35%
solid residue 5%

Other data:

heat of vaporization of gasolines $H_{vg}$ = 243.22 kJ/kg (at 200 °C)
heat of vaporization of oils $H_{vo}$ = 214.51 kJ/kg (at 300 °C)
heat of vaporization of paraffins $H_{vp}$ = 196.93 kJ/kg (at 350 °C)
cracking heat $H_{crac}$ = 1465 kJ/kg

The heat demand of the reactor determined for the given parameters is:

$$Q_R = Q_1 + Q_{2g} + Q_{2o} + Q_{2p} = 434.60 \text{ kW} + 5.07 \text{ kW} + 340.00 \text{ kW} + 14.36 \text{ kW} = 779.67 \text{ kW} \approx 800 \text{ kW}$$

The amount of heat calculated above is not the amount of real heat that needs to be supplied to the reactor to run the process. To assess the real value of heat demand, we should additionally take into account in calculation heat losses through the insulation ($Q_i$—approximately 10%) and the heat carried with the flue gas ($Q_{fg}$—approximately 40%).
The total amount of heat supplied to the reactor will be then:

$$Q_{TotR} = Q_R + Q_i + Q_{fg} = 800 + 80 + 320 = 1200 \text{ kW} \ (4{,}320{,}000 \text{ kJ})$$

4.1.3. The Oil Fraction Produced in the Process of Thermal Waste Treatment

In the pyrolysis process, the sorted waste stream is transformed into liquid fuel. Table 5 lists selected parameters characterizing the oil fraction obtained in the technological process designed by the plant with the requirements for standard fuel oil contained in the Regulation of the Minister of Economy of 9 October 2015 on quality requirements for liquid fuels [52].

The results are presented for four samples collected during 18 months of reactor operation. Samples were taken every four or five months, allowing characterization of the liquid fraction obtained by pyrolysis of waste derived from different sources of supply.

As you can see from Table 5, currently all liquid fractions obtained in the pyrolytic process do not meet the requirements in regard to parameters 3 and 6. Also, most fractions do not meet the requirements in regard to parameter 4. Requirements for parameters 1 and 7 are not fully fulfilled for one of the tested samples. The results achieved indicate that obtained oil fractions can currently be used alternatively as excise energy products or hydrocarbon components for further processing in industry. These products are now used as additives to fuel oils and other products of refinery origin as well as a number of products of common usage such as adhesives and solvents. In future, in order to be able to convert them into liquid fuels of a quality not different from that of conventional fuel oils,

they need to undergo a treatment process that will increase the flash point temperature of liquid fractions and limit sulfur and contamination contents in them.

**Table 5.** Values of selected parameters characterizing oil fraction obtained in the technological process designed by the plant together with requirements for standard fuel oil.

| No. | Medium and Heavy Fraction | | Method applied | Sample1 | Sample2 | Sample3 | Sample4 | min | max |
|---|---|---|---|---|---|---|---|---|---|
| | Cetane index | | | | $53.45 \pm 0.25$ | | | 46 | |
| 1 | Density at 15 °C | kg/m$^3$ | PN-EN ISO 12185:2002 | 830.5 | 842.5 | 833.2 | 813.5 | 820 | 845 |
| 2 | Polycyclic aromatic hydrocarbons content | % (m/m) | PN-EN 12916:2016-03 | 0.08 | 4.7 | 3.7 | 3.4 | | 8 |
| 3 | Sulfur content | mg/kg | PN-EN ISO 14596:2009 | 278 | 29 | 20 | 63 | | 10 |
| 4 | Flash point | °C | PN-EN ISO 2719:2016-08 | <40 | 58.5 | 45.5 | 49 | >55 | |
| 5 | Water content | mg/kg | PN-EN ISO 12937:2005 | 110 | 40 | 130 | 150 | | 200 |
| 6 | Impurity content | mg/kg | PN-EN 12662:2014-05 | >30 | >30 | >30 | >30 | | 24 |
| 7 | Viscosity at 40 °C | mm$^2$/s | PN-EN ISO 3104:2004 | 1.692 | lack of data | 2.622 | 2.764 | 2 | 4.5 |
| 8 | Distills up to 250 °C | % (*v/v*) | | 56.1 | 10.00 | 33.1 | 58.9 | | <65 |
| 9 | Distills up to 350 °C | % (*v/v*) | PN-EN ISO 3405:2012 | 91.1 | 98.3 | 97.9 | 97.6 | 85 | |
| 10 | Distills 95% (*v/v*) to temperature | °C | | * | 344.5 | 333 | 325.9 | | 360 |

* 94% (*v/v*) has been distilled up to 370 °C.

## 5. Conclusions

The main assumption of the technology on which the operations of the GreenTech Polska S.A. plant are based is the recovery of energy accumulated in polymeric materials from the group of polyolefins of waste origin. The development of this type of technology perfectly fits into the area of the circular economy and renewable energy sources, therefore it is a response to the ever-growing problem of ensuring energy and economic security for countries. The sorting solutions used in the developed technological process allow for efficient separation of the inert (glass, metals, paper, etc.) and for the separation of the plastics subject to further pyrolysis. The processing capacity of the designed installation is 1000 kg/h, which, assuming 8000 working hours per year, ensures the processing of approximately 8000 tons of plastic waste per year. The efficiency of the pyrolysis module is 80–90% of the raw material weight, which means that from 1 ton of sorted plastics, 800–900 kg of liquid fraction can be obtained. The liquid product is a mixture of paraffinic hydrocarbons and aliphatic cyclohexane derivatives (naphthenes) and olefins with carbon atoms in the C6–C25 molecule, boiling range 35–370 °C and freezing point −30 to +40 °C. The composition of the final product depends on the composition of the raw material used (the percentage of polyethylene (PE), polypropylene (PP), and polystyrene (PS) in mixture). Depending on parameters such as flash point, sulfur content, density, viscosity,

and so forth, liquid fractions obtained by pyrolysis can be used as fuels and fuel oils or as additives to heating oils and other products of refinery origin as well as a range of common products such as adhesives or solvents.

The conducted research has shown that although the technological process, which is the basis of GreenTech Polska S.A. activity, is relatively simple to carry out under laboratory conditions, its application on an industrial scale poses a major engineering challenge. Although the design solutions introduced by the GreenTech Polska S.A. company allow efficiency from a technological and economic point of view (the average company income per year is 1,246,760.00 EUR [53]), the pyrolysis process, like any new technology, should be subject to further improvement. The advantage of the designed technology is the possibility to carry out the pyrolysis at relatively low temperatures and without the use of a catalyst, however, the limitation of the developed process is manufacturing low-quality fuel products as indicated by such parameters as sulfur content, flash point temperature, and impurity content of liquid fractions. This problem could be solved by additional cleaning of the sorted reaction mass from solid particles and extending the technological line with a distillation column. The recommended direction of improvement of the technology is also optimization of the process of the reactor's purification and disposal of contaminants (improving operation of the filters, introduction of a multistage system with possibility of switching off any stage, adding a hydrocyclone at first stage and/or sedimentation tank (e.g., Dorr tank)).

The above-mentioned actions will enable more effective purification and isolation of the fractions, which in turn will increase the quality of obtained fuels.

**Author Contributions:** Conceptualization, E.S.; methodology, E.S.; validation, N.G., P.D. and A.D.; formal analysis, E.S.; investigation, E.S.; resources, A.F. and P.M.; data curation, A.F. and P.M.; writing—original draft preparation, E.S.; writing—review and editing, E.S., N.G. and P.D.; visualization, E.S.; supervision, E.S. All authors have read and agreed to the published version of the manuscript.

**Funding:** This work was supported by the "Program Operacyjny—Innowacyjna Gospodarka Działanie 4.4." program, which is gratefully acknowledged.

**Institutional Review Board Statement:** Not applicable.

**Informed Consent Statement:** Not applicable.

**Data Availability Statement:** The data presented in this study are available on request from the corresponding author. The data are not publicly available due to company protection policy.

**Conflicts of Interest:** The authors declare that they have no known competing financial interests or personal relationships that could have appeared to influence the work reported in this paper. Any documentation, certificates, attestations, approvals, etc., referred to in the paper and marked with characteristics such as number, mark and unique name are in the possession of the Plant Manager and are available for inspection.

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
