# Peer review of "Characteristics of Plastic Waste Processing in the Modern Recycling Plant Operating in Poland"

_energies, doi:10.3390/en14010035_

Round 1

Reviewer 1 Report

The authors provide an interesting and detailed description of the GreenTech Polska S.A. plant and analyze the input and output streams in terms of their composition and ISO fuel standards. However, the description is at times a bit unclear and unspecific. In addition, no clear final conclusion is reached and the abstract has to be improved. Some general remarks below, followed by specific pointers to how the paper can be improved.

Typically depolymerization refers to solvolysis type processes, where monomers can be obtained as the main product. I recommend to use pyrolysis or thermal conversion exclusively in the paper. (lines 18, 97, 116, 121, 124, 125, 127, 129, 137, 177, 226, 227, 230, 248, 351, 358)

The authors should take care in the use of articles. A lot of times these are missing or are specific “the” while they should be unspecific “a”. For the example “the calorific value exceeding 6 MF kg-1” in line 304.

  • Line 322: the oil fraction
  • Line 48: a fight
  • Line 51: a recycling process
  • Line 52: a regulation

The authors have to improve the abstract. It lacks specific information. What aspects did the authors exactly analyze. I see that the authors analyzed the composition of the input and output of the process and compared it to required fuel specifications, but what about energy requirements and consumption for example? Then the authors mention specific parameters that should be taken into account to improve the designed process. I want to read, what those parameters are already in the abstract. Actually, still after reading the paper it is not clear to me, what those parameters are. Also in line 18: I suggest to be more specific. I.e. “pyrolysis of polyolefin packaging waste”

Line 62: Do the authors mean the waste coming from within (of Poland) and from outside of Poland (of the EU). I suggest replacing “of” by “from”

Line 70: number instead of indicator, an indicator can be anything

Line 71: One of the highest in the whole of Europe

Line 100: I suggest specifying what the “technological purposes” are. It is later mentioned that propane is used for heating during startup of the plant.

Line 108: mechanical processing can be confused with mechanical recycling via melting and re-extrusion. What the authors mean is shredding and sorting and I recommend calling it that way to be more specific.

Line 115: How is the waste separated? Is it sorted by hand?

Line 117: What do the authors mean by “bonded” Is the waste molten together? Do the authors mean that contact homogenization is used for bonding? In that case what is contact homogenization. Can the authors explain?

Line 120: How is it ensured that only these types of polymers are separated? How are the polymer types identified? IR sorting?

Line 140: I disagree that pyrolysis leads to a high value product. High value compared to what? Other recycling products yield a more pure and thus valuable product, i.e. glycolysis of PET.

Line 150: In relation to what are CO2 emissions decreased. CO2 emissions of pyrolysis are higher than those of mechanical recycling. They are only lower in comparison to incineration (https://onlinelibrary.wiley.com/doi/full/10.1002/anie.201915651)

Figure 5, second box from top on the right: Typically crosslinking does not yield a more volatile product.

Figure 5, last box from top on the right: Not only side chains can cyclize, but this is what the authors imply.

Line 159: I think the authors mean propagation not depropagation. Do the authors mean radical propagation? It seems like it from the description that follows.

Line 174: How can a recombination of two molecules lead to a molecule that has a lower molecular weight than each of the pieces that combine.

Line 176: I suggest to use higher carbon to hydrogen ratio instead of percentage.

Line 235: The authors should describe the purpose of the extractor and how it works. What is the solvent feed mechanism. Where does the solvent come from, the reactor? And where does it go? What is the solvent needed for? Is the heavy fraction from the reactor still liquid? How does it get from the reactor to the extractor. Is this heavy fraction the solvent?

Line 237: What is the purpose of the homogenization process? What is homogenized?

Line 259: It is confusing that the authors start this explanation with the last step and not with the first one. Also it would help if the authors would number the units in the flow scheme (Figure 7) and refer to them in the text.

Line 284: What is meant by “the substrate of pyrolysis”? The input to the pyrolysis unit?

Line 299: What is meant by “fine mixed waste”? Waste that is pulverized? It is mentioned later as fine fraction <10 mm, why not mention this earlier to make it easier to follow?

Line 306: “landfilled in the landfills” seems double

Line 308: by the “reject fraction” the authors mean the reject from mechanical sorting or from the pyrolysis unit?

Line 309: “or [in] waste incinerators”

Line 315: transferring instead of transfer. Does this mean, this fuel stream is incinerated together with other waste or it is used to fuel the incinerator.

Line 316: This sentence is not clear to me. What kind of renewable source, a source for what? How and to what purpose should you include “this procedure” in the renewable source. What kind of impact will it have on the values of carbon dioxide emissions.

Line 349: What kind of challenges? I did not read a discussion of those in the paper. There is also no discussion of why a certain design was chosen.

Author Response

Dear Reviewer,

We would like to thank you for your valuable comments and suggestions, which helped us to improve the quality of the article.
Below please find a point-by-point response to all comments. All the comments were taken into consideration and relevant changes were made in the text. The corrections and fragments added following the comments are marked in yellow.

Reviewer 2 Report

The manuscript named by "Characteristics of the technological process and plastic processing line on the example of a plastic waste recycling plant operating in Poland." includes important content and fullfill scientific paper attributes. Especially in Poland, this kind of information and studies with waste recycling is needed. However, I still waiting some corrections before publication.

Title it is a boring and it could be more informative. Point (.) at the end of title is useless. Consider to rewrite title.

Abstract. Open to abbreviation EU after the first mention. SEE LINE 17 "...14 countries of UE". #!#!# student mistake

Figure 1. Resolution is poor, can you fix it?

Figure 2. What is the informative level of this figure? I could understand this figure if the size of countries have been presented according to the number in parentheses, eg. Germany size could be ten times higher compared to Austria... Now, this figure looks like a flag presentation for a kindergarden...

2.1.1. Production plant. This section must be reformulated before publication, eg. by using indent... such as line 232 onward.

Table 1 includes "Waste code". How has determined these codes? which standard?

Discussion about the results could be deeper...

References: Some names for internet sources could be useful, references 1 and 2, and 6-9, as well as 52. Some "more scientific information" could be better for these references... search it!

Author Response

Dear Reviewer,

We would like to thank you for your careful and thoughtful comments about our manuscript. We really appreciate the time and effort that have gone into your detailed comments. These comments are all valuable and very helpful for revising and improving our manuscript, as well as the important guiding significance to our researches.
Below please find a point-by-point response to all comments. All the comments were taken into consideration and relevant changes were made in the text. The corrections and fragments added following the comments are marked in green.
